# Impact of depression on stroke outcomes among stroke survivors: Systematic review and meta-analysis

Seble Shewangizaw[1]*, Wubalem Fekadu[1], Yohannes Gebregzihabhier[1,2], Awoke Mihretu[1], Catherine Sackley[3], Atalay Alem[1]

1 WHO Collaborating Centre for Mental Health Research and Capacity Building, Department of Psychiatry, College of Health Sciences, Addis Ababa University, Addis Ababa, Ethiopia, 2 Department of Nursing, Debre Berhan University, Debre Berhan, Ethiopia, 3 Faculty of Medicine and Health Sciences, University of Nottingham, Nottingham, United Kingdom

* sebleshewangizaw@yahoo.com

## Abstract

### Background

Depression may negatively affect stroke outcomes and the progress of recovery. However, there is a lack of updated comprehensive evidence to inform clinical practice and directions of future studies. In this review, we report the multidimensional impact of depression on stroke outcomes.

### Methods

**Data sources.** PubMed, PsycINFO, EMBASE, and Global Index Medicus were searched from the date of inception.

**Eligibility criteria.** Prospective studies which investigated the impact of depression on stroke outcomes (cognition, returning to work, quality of life, functioning, and survival) were included.

**Data extraction.** Two authors extracted data independently and solved the difference with a third reviewer using an extraction tool developed prior. The extraction tool included sample size, measurement, duration of follow-up, stroke outcomes, statistical analysis, and predictors outcomes.

**Risk of bias.** We used Effective Public Health Practice Project (EPHPP) to assess the quality of the included studies.

### Results

Eighty prospective studies were included in the review. These studies investigated the impact of depression on the ability to return to work (n = 4), quality of life (n = 12), cognitive impairment (n = 5), functioning (n = 43), and mortality (n = 24) where a study may report on more than one outcome. Though there were inconsistencies, the evidence reported that depression had negative consequences on returning to work, functioning, quality of life, and mortality rate. However, the impact on cognition was not conclusive. In the meta-analysis,

**Data Availability Statement:** All relevant data are within the manuscript and its Supporting Information files.

**Funding:** This work was supported through the DELTAS Africa Initiative (DEL-15-01) by funds awarded to SS. The DELTAS Africa Initiative is an independent funding scheme of the African Academy of Sciences (AAS) Alliance for Accelerating Excellence in Science in Africa and supported by the New Partnership for Africa's Development Planning and Coordinating Agency (NEPAD Agency) with funding from the Wellcome Trust (DEL-15-01) and the UK government. The views expressed in this publication are those of the author(s) and not necessarily those of AAS, NEPAD Agency, WellcomeTrust or the UK government. There was no additional external funding received for this study. The funders mentioned had no role in study design, data collection and analysis, decision to publish, or preparation of the manuscript.

**Competing interests:** The authors have declared that no competing interests exist.

depression was associated with premature mortality (HR: 1.61 (95% CI; 1.33, 1.96)), and worse functioning (OR: 1.64 (95% CI; 1.36, 1.99)).

## Conclusion

Depression affects many aspects of stroke outcomes including survival The evidence is not conclusive on cognition and there was a lack of evidence in low-income settings. The results showed the need for early diagnosis and intervention of depression after stroke. The protocol was pre-registered on the International Prospective Register of Systematic Review (PROSPERO) (CRD42021230579).

## Introduction

Stroke is a neurological deficit attributed to an acute focal injury of the central nervous system by a vascular cause including cerebral infarction, intracerebral and subarachnoid hemorrhage [1]. They often face a range of problems including the inability to move some or whole parts of the body, problems with bladder and bowel control, numbness or strange sensations, trouble with judgment and memory, problems of understanding or forming speech, trouble in controlling or expressing emotions and, experiencing depressive symptoms [2].

Depression is one of the most common neuropsychiatric disorders that can happen before or in the early or late stages of a stroke. It affects approximately one-third of stroke survivors [3]. Post-stroke Depression (PSD) can occur as a continuation of pre-existing depression or may develop after the stroke. PSD is related to poor functional outcomes [5] and is associated with an increased mortality risk [4].

Kutlubaev and Hackett [5] conducted a systematic review of the predictors of depression after a stroke and the impact of depression on stroke outcomes. They reported a negative association between functional outcomes and PSD. Bartoli et al. [6, 7] and Cai et al. [8] also conducted a review and reported an increased mortality rate among survivors with depressive symptoms. Blöchl et al. [9] conducted a review on the impact of PSD on physical disability and reported poor functional outcomes among survivors with depressive symptoms.

Though these reviews included important studies and reported the impact of depressive symptoms on stroke outcomes they fail to review the other dimensions of stroke outcomes such as the ability to return to work. Our review included more studies and assessed more dimensions of stroke outcomes (the ability to return to work, cognition, and quality of life). Therefore, this systematic review and meta-analysis aimed to examine the relationship between depression and stroke outcomes (returning to work, functional recovery, cognition, quality of life, and mortality rate).

## Methods

We were guided by the Preferred Reporting Items for Systematic Reviews and Meta-Analysis (PRISMA) [10] guidelines to report the review. The protocol was pre-registered on the International Prospective Register of Systematic Review (PROSPERO) (CRD42021230579).

### Search strategy

We searched four databases from the date of inception until $1^{st}$ August 2023: PubMed, Embase, Global Index Medicus, and PsycINFO. Forward and backward search was conducted

for the included studies. We have also searched University repositories and Google Scholar for grey literature. We used three big terms (terms for stroke, terms for depression, and terms for outcome) which were combined by the Boolean term AND (S1 File).

### Eligibility criteria

We included longitudinal studies on depression conducted among adults diagnosed with various types of strokes. We included studies that reported both clinically diagnosed depression and studies that utilized screening tools. The outcomes we looked at were functioning, quality of life, returning to work, cognition, and mortality (rate and premature mortality)

### Study selection process

The identified references were exported into EndNote reference manager software [11] and duplicates were removed. The references were reviewed using their title and abstracts. After that, the full body of the selected articles was checked for inclusion criteria. We then extracted the author, year of publication, sample size, measures used, and results of the articles. The article screening, selection, and extraction were done by two independent investigators (SS and WF). A third reviewer resolved discrepancies (YG).

### Quality assessment

The quality of included studies was evaluated by the two investigators (SS and WF) independently using Effective Public Health Practice Project (EPHPP) [12]. EPHPP provides the means to assess study quality using its eight sections which include selection bias, study design, confounders, blinding, data collection methods, withdrawals and dropouts, intervention integrity, and analysis. Results lead to an overall methodological rating as strong, moderate, or weak.

### Data synthesis

We conducted a narrative synthesis to report the impact of depression on the ability to return to work, cognition, quality of life, functioning, and mortality rate. In the meta-analysis (homogeneous studies), we reported the pooled impact of PSD on functioning and mortality rate. For the mortality rate, we reported two pooled estimates as some studies reported hazard ratio (HR) while others reported odds ratio (OR). Since we expected heterogenicity we conducted a random effect meta-analysis. We used a funnel plot to see the risk of publication bias. We also conducted heterogeneity tests ($I^2$) to examine the variation in the outcomes among the studies. We used Comprehensive Meta-analysis Software 4 for the meta-analysis [13].

### Patient and public involvement

No patient was involved.

## Results

### Study selection

Initially, we identified 25,646 articles. After removing 5,996 duplicates, 19,652 articles were screened using their title and abstract. Then, 150 full articles were reviewed, and we excluded studies which did not fulfil the inclusion criteria. Finally, eighty prospective studies that reported the relationship between depression and one or more stroke outcomes of interest

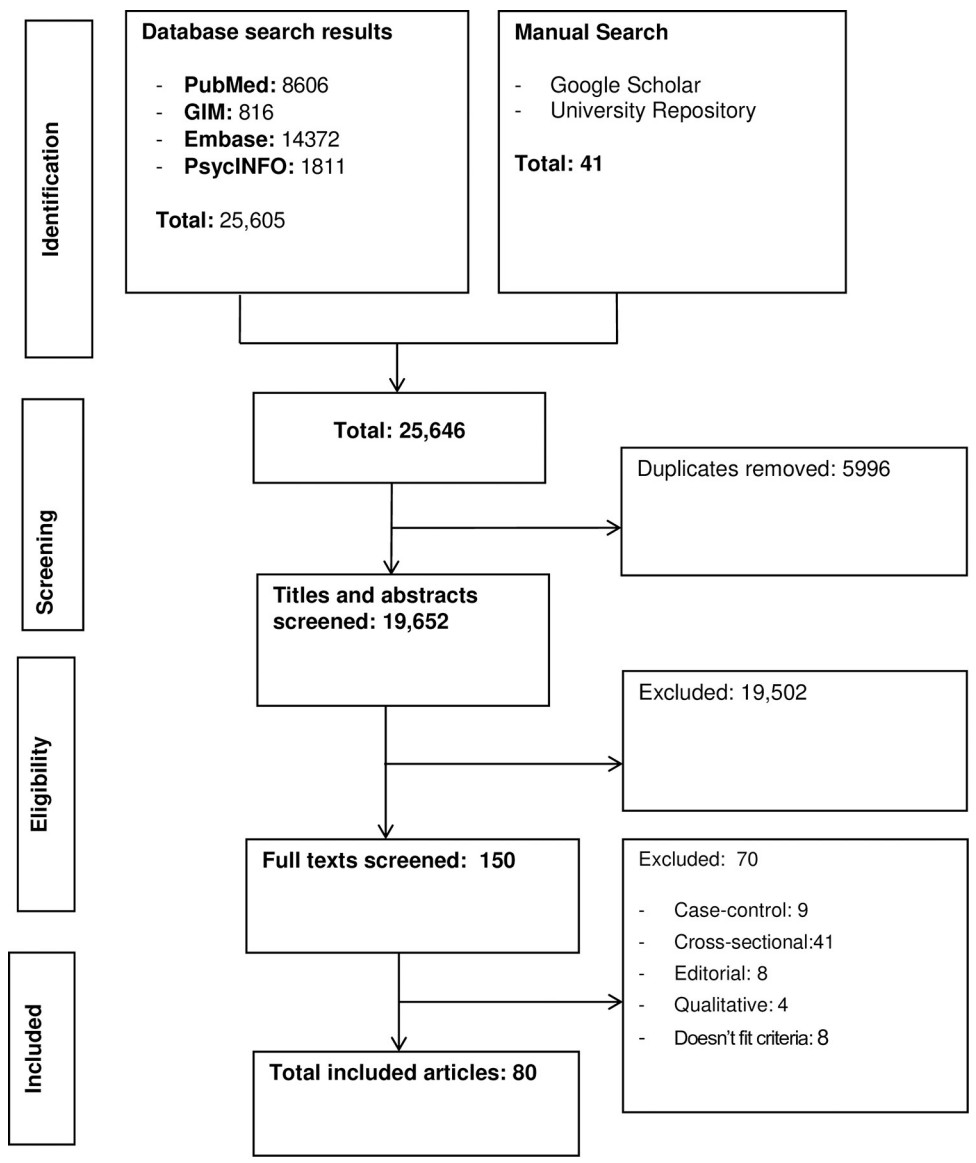

**Fig 1. PRISMA flow diagram of the study selection process.**

(cognition, returning to work, quality of life, functional recovery, and mortality rate) were included (Fig 1).

## Characteristics of the studies

Most of the studies (n = 66) were from high-income countries (HIC) (England, USA, Denmark, Norway, Korea, Australia, Canada, Poland, Finland, Singapore, Italy, Germany, Ireland, The Netherlands, Saudi Arabia, Japan, Scotland, and Singapore); and the rest (n = 14) were from middle-income countries (China, Nigeria, Brazil, Tunisia, Thailand, Lebanon, India and Serbia) and we did not come across studies from low-income countries which fulfilled the eligibility criteria.

Out of the eighty studies included in this review, four studies investigated the impact of depression on the ability to return to work, five on cognitive impairment, twelve on quality of

life, forty-three on functional recovery and twenty-four on mortality. The number of participants in the reviewed studies ranged from 49–152, 243 at baseline assessment.

## Risk of bias within the studies

The global rating of the articles based on the EPHPPs resulted in thirteen strong, fifty-six moderate, and eleven weak quality studies. The papers that were categorized as weak lacked an assessment of different factors that may affect the main outcome (S2 File).

## Impact of depression on returning to work (n = 4)

Four studies [14–17] assessed factors predicting the ability to return to work after a stroke. These studies were from Australia [14], Brazil [15], the United States of America (USA) [16], and the Netherlands [17]. The mean age of the participants was 50 years. They considered time up to one month from the stroke incident as a baseline and followed them up to two years (Table 1).

Stroke survivors who had worked for at least a month before the stroke incident were eligible to participate. Different types of professions were included from full-time to part-time jobs with different occupations including self-employed and non-manual jobs.

Out of the four studies, one study reported that depression affected the ability to return to work [14] while the other three reported no significant association between returning to work and depression. This study reported higher odds of returning to work within six months among survivors without depressive symptoms (AOR = 4.92, 95% CI, (1.92–14.37)) [18]. None of the studies reported treatment for depression (Table 1).

## Impact of depression on quality of life (n = 12)

Twelve studies [19–30] reported the impact of depression on stroke survivors' quality of life; these studies were from the UK [19], Australia [30], Tunisia [20], Ireland [22], Nigeria [23], Brazil [24], Korea [21, 25], the Netherlands [26], China [27, 29], and Serbia [28]. The age range of the participants was 20–98 years; the sample size ranged from 49–1, 101 participants.

Ten studies found that depressive symptoms at various stages of recovery could affect the quality of life of a stroke survivor. The impact was true on both physical and mental components of quality of life.

In two studies that reported no association between depression and quality of life; one study reported that only 4% of the variance in the quality of life score was explained by depression while other predictors like economic status predicted 12% and activities of daily living predicted 19% of the variance [31] (Table 1).

## Impact of depression on cognition (n = 5)

Studies conducted in England [19], Brazil [32], Finland [33], Lebanon [34] and South Korea [35] investigated the relationship between PSD and cognitive impairment. Baseline measures were taken one month after the stroke and followed up for 6 months in the South Korean study, and baseline measures were taken three months after the stroke and followed up for five years in the England study. Stroke survivors with severe cognitive impairment [19] or communication impairment due to dysphasia or dysarthria [19, 33, 35] and previous major depression with a history of suicide [32] were excluded from these studies.

Three of the studies reported no significant relationship between depression and cognitive impairment. Depression at 3 months was not associated with cognitive impairment at any point in time throughout the 5 follow-up years. Even though the mean Global deterioration

**Table 1. Impact of depression on quality of life and returning to work.**

| Author and year | Setting | Sample size | Age | Gender | Follow up | Depression Measures | Outcome measure | Depression treatment | Results |
|---|---|---|---|---|---|---|---|---|---|
| Ayerbe et al. 2014 | England | 1101 | 0-64- 35.7% >64– 64.3% | Male 54% | Up to 10 years | HADS | SF-36 | Not reported | • Depression at 3 months and QOL in 5 years. ○ $B$ = -8.16 (- 10.23, 6.15) • Depression at 5 years and QOL in 10 years. ○ $B$ = -8.16 (- 10.23, 6.15) |
| Boudokhane et al. 2021 | Tunisia | 49 | Mean 62.2 | Male 60% | Up to 1 year | SF-36 | HADS | Not reported | • Depression a 1 month and QOL at one year • B = -0.54 (-1.55,0.72) $p<0.001$ |
| Donnellan et al. 2010 | Ireland | 107 | Range 20–98 | Male 51% | Up to 1 year | SSQOL | HADS | Patients receiving treatment were excluded from the study | • Depressive symptoms and poorer quality of life ○ T1 (r = -0.56, $p<0.001$) and at T2 (r = -0.41, $p<0.001$) |
| Gbiri et al. 2010 | Nigeria | 65 | Mean 58.1 | Male 33 | Up to 6 months | SSQOL | ICD-10 | Not reported | • Depression at baseline and QOL at 3 months ○ $R^2$ = 0.31 p = 0.00 • Depression at baseline and QOL at 6 months ○ $R^2$ = 0.53 p = 0.00 |
| Guajardo et al. 2015 | Brazil | 75 | Mean 51.6 | Male 66.7% | Up to 3 months | GDS | BDI | Not reported | • Depressive and SF-36 subscale vitality ○ R = - 0.319, $P<0.01$ • Depressive and SF-36 subscale mental health ○ R = -0.257, $P<0.05$ |
| Hackett et al. 2012 | Australia | 271 | Range 17–65 | Male 68% | 28 days, 6 and 12 months | HADS | - | Not reported | • Depression and returning to work. ○ OR = 2.31 (0.87, 6.12) |
| Jet van der Kemp et al. 2019 | Netherland | 121 | Mean 56.3 | Female 27.3% | 2 months and 1 year | HADS | - | Not reported | • Depression not predicting returning to work by 1-year post-stroke. ○ $B$ = -0.094 P = 0.094 |
| Nascimento et al. 2019 | Brazil | 117 | Mean 57 | Male 68% | 3 months 6 months | HADS | - | Not reported | • Depression did not predict returning to work |
| Kim et al. 2018 | Korea | 423 | Mean 64.5 | Male 57.7 5 | Up to 1 year | WHOQOL | MINI | Depression was not treated. | • Depression had a significant and persistent impact on QOL at 2 weeks and 1 year after stroke. |
| Lam et al. 2019 | Netherland | 120 | Mean 68.6 | Male 64.2% | Up to 1 year | RAND-36 | HADS | Not reported | • Depression at baseline and QOL at one year ○ $B$ = -1.35, $p<0.001$ |
| Li et al. 2019 | Beijing, China | 801 | Mean 57.5 | Female 30.6% | Up to 5 years | SF-12 | DSM-IV | Not reported | • Persistent depression at one year and poor MSC score at 5 years. ○ OR = 48, (0.29,0.81) |
| Orman et al. 2022 | Australia | 563 | Mean 68.4 | Male 64.5% | Up to 2 years | AQoL-4D | HADS | Not reported | • Depression associated with lower AQOL-4 scores. • ß = -0.058(-0.11,0.00), p = 0.05 |
| Schulz et al. 2017 | USA | 159 | Range 40–86 | Male 74.8% | 3,6, 9 and 12 months | GDS-15 | - | Not reported | • Depression was not a predictor of SS's ability to return to work |
| Shi et al. 2016 | China | 747 | Mean 61 | Female 32.1% | Up to 1 year | SF-36 | HDRS | Patients with antidepressant treatment were excluded from the study. | • Depression and PCS ○ OR = 0.43 (0.30,0.62) • Depression and MCS ○ OR = 0.33 (0.23,0.47) |

*(Continued)*

**Table 1.** (Continued)

| Author and year | Setting | Sample size | Age | Gender | Follow up | Depression Measures | Outcome measure | Depression treatment | Results |
|---|---|---|---|---|---|---|---|---|---|
| Smi et al. 2006 | Korea | 214 | Mean 63 | Male 61% | Up to 3 years | WHOQOL | DSM-IV | Not reported | • Depression and QOL ○ $R^2$ = 0.04 P< 0.05 |
| Zikic et al. 2014 | Serbia | 60 | | | Up to 6 weeks | SF-36 | HDRS | Not reported | • SF-36 mean scores were higher in patients without depression |

AQoL-4D - Assessment of Quality of Life instrument, CI–Confidence Interval, MCS- Mental Component Summary, OR–Odds Ratio, PCS–Physical Component Summary, QOL–Quality Of Life, DSM-IV–Diagnostic Statistical Measure IV, HADS -Hospital Anxiety and Depression Scale, HDRS-Hamilton Depression Rating Scale, ICD-10- International Classification of Disease, MCS- Mental Component Summary, MINI–the Mini International Neuropsychiatric Interview, PCS–Physical Component Summary, PSD–Post-Stroke Depression, RAND-36 –Research and Development 36 Scale, SF-36 –the Short Form 36 health survey questionnaire, SS-Stroke Survivor, SSQOL- Stroke Specific Quality Of Life, QOL- Quality Of Life WHOQOL- World Health Organization's Quality Of Life Measure

scale (GDS) score was lower among the mild to moderate depression (MMD) group compared to the moderate to severe depression (MSD) group; there was no significant difference in the change of GDS scores over time. While Finland [33], and Lebanon study [34], reported a significant association between depression and the degree of cognitive deficit (Table 2).

## Impact of depression on functional recovery (n = 43)

Different terminologies such as activities of daily living, dependency, functioning, and motor function were used. Forty-three studies [19, 22, 28, 29, 35–73] reported the impact of depression on functional recovery. All except two studies from Serbia [28] and China [29, 72] (upper middle-income countries) were from high-income countries. The number of participants ranged from 40–1753 (Table 2).

Thirty-two studies reported a significant association between depression and functional outcomes. Depression at baseline predicted functioning after 6–24 months (OR:2.7–3.7). The severity of depression was also associated with poor outcomes at 6 months and one year after the stroke. Stroke survivors with MSD had poorer outcomes compared to those who were MMD [29, 35, 69].

Stroke survivors with depressive symptoms had a lower score on a motor assessment scale. For every point increase on the depressive symptom scale, there was a decrease of 0.82 points and 0.77 points on different motor outcome scales indicating poorer outcomes [50]. The crucial part of stroke rehabilitation, trunk control at discharge was also influenced by the presence of depression at admission into the rehabilitation center ($B$ = 9.057 (1.03,17.08) p = 0.027) [59] (Table 2).

On the other hand, 11 studies reported depression not related to functional outcomes even though one study reported PSD predicting inactive lifestyle rather than functional outcome or performance [39]. And in a study conducted in Norway found that depression was not an independent predictor for modified Rankin scores [68]. In a study conducted in India, no significant difference in the functional outcomes between stroke patients with depression and those without depression with inpatient rehabilitation programs was reported [73].

Regarding depression treatment, out of the total 43 studies, 15 studies [38, 40, 43, 47, 51–54, 56, 57, 60, 62, 67, 72, 73] reported the number of participants who were on treatment for depression whether it was anti-depressant or psychological therapy while one study excluded patients who were on treatment for depression [22]. In a few of these studies [38, 51, 67], functioning improvement was not associated with treatment for depressive symptoms (Table 2).

**Table 2. Impact of depression on cognition and functional recovery.**

| Author, year | Setting | Sample size | Follow up | Depression measures | Outcome measure | Depression treatment | Results |
|---|---|---|---|---|---|---|---|
| Adbdul-Sattar et al. 2013 | Saudi Arabia | 180 | Up to 28 days | GDS-15 | FIM | Not reported | • Presence of depression was negatively associated with FIM scores. ○ *Beta = -3.73 SD = 0.85* |
| Ayerbe et al. 2014 | England | 1101 | Up to 5 years | HADS | BI | Not reported | • Depression at 3 months and disability at 5 years ○ RR = 4.71 (2.96, 7.48) |
| Ayerbe et al. 2015 | England | 1307 | Up to 3 years | HADS | BI | Not reported | • Depression at 3 months and disability at 3 years ○ RR = 4.01 (2.42–6.63) P < 0.001 |
| Baccaro et al. 2019 | Brazil | 103 | Up to 6 months | DSM IV | MMSE MoCA | Not reported | • Depression symptoms were not associated with cognition scores |
| Boutros et al. 2023 | Lebanon | 150 | | HADS | MMSE | Not reported | • Depression and cognitive impairment • AOR = 2.536, CI = [1.004–6.403], p = .049 |
| Cassidy et al. 2004 | Ireland | 50 | Up to 2 months | HDRS | BI | 6 patients were on antidepressant | • Depression was not related to functional disability. |
| Clark et al. 1998 | Australia | 125 | Up to 1 year | ZSRS Australian | ADL | Not reported | • Depression was not related to functional status. • Depression is strongly a negative predictor of an inactive lifestyle at 6 and 12 months |
| Donnellan et al. 2010 | Ireland | 107 | Up to 1 year | HADS | Nottingham Extended ADL scale | Patients receiving treatment were excluded from the studies. | • Depressive symptoms and poorer functional ability at • At T1 r = -0.29, p<0.01 & T2 at one year r = -0.19, p<0.001 |
| El Husseini et al. 2017 | USA | 1444 | Up to 1 year | PHQ-8 | MRS | 18.2% of participants were on antidepressant | • Persistent depression and worsening MRS • OR = 0.85 (0.53,1.34) |
| Gillen et al. 2001 | USA | 243 | Up to 1 month | GDS | FIM | Not reported | • Higher level of depressive symptoms affects recovery outcome. |
| Gupta et al. 2022 | India | 30 | Up to a year | HADS | BI, MRS | 9 patients received antidepressant | • No significant difference in functional outcome between survivors with depression and without depression |
| Hama et al. 2007 | Japan | 237 | Up to 5 months | SDS | FIM | Not reported | • SDS scores did not predict FIM scores |
| Herrmann et al. 1998 | Canada | 436 | Up to 1 year | MADRS | FIM | 19% of depressed patients were on antidepressant | • Higher depression scores and functional outcome ○ $R^2$ = 0.39, p< 0.0001 |
| Johnston et al. 2004 | Scotland | 40 | Up to 3 years | HADS | BI | Not reported | • HADS depression score was not a significant predictor of recovery. |
| Kang et al. 2018 | Korea | 145 | Up to 1 year | MINI | MRS | Not reported | • Depression at the acute phase predicted poor functional outcomes during both the acute and chronic phases of stroke |
| Kauhanen et al 1999 | Finland | 106 | Up to 1 year | DSM III R | MMSE | 19/53 depressed patients used antidepressant | • significant association between the categories of depressive illness and the degree of cognitive deficits |
| Kijowski et al. 2014 | Poland | 423 | Over a year | GDS | MRS | Not reported | • Depression limits gait recovery after stroke. |
| Koivisto et al. 1993 | Finland | 143 | Up to 2 ½ months | DSM-III-R | MAS of Sivenius | Not reported | • SSs with depression performed worse compared to non-depressed SSs in ADL at follow up |
| Kotila et al. 1999 | Finland | 523 | Up to 1 year | BDI | BI | 18% of depressed patients were on antidepressant | • Depression at 3 months was associated with poor functional outcomes at 12 months. |
| Kuptniratsaikul et al. 2009 | Thailand | 271 | Up to 1 month | HADS | BI | Not reported | • Depression was not a predictor of BI scores. |
| Lai et al. 2002 | USA | 459 | Up to 6 months | GDS | BI | Not reported | • Depression and BADL • Risk ratio = 0.3 (0.23,0.50) |

(*Continued*)

**Table 2.** (Continued)

| Author, year | Setting | Sample size | Follow up | Depression measures | Outcome measure | Depression treatment | Results |
|---|---|---|---|---|---|---|---|
| Lin et al. 2020 | USA | 57 | Up to 3 months | PHQ-9 | MRS | Not reported | • Higher PHQ-9 scores were associated with worse motor outcome |
| Loong et al. 1995 | Singapore | 52 | Up to 1 month | HDRS | MBS | 10 patients were on antidepressant | • Depressive during admission were associated with functional impairment at discharge. |
| Matsuzaki et al. 2015 | Japan | 117 | Up to 2 months and 20 days | SDS | FIM | 10 patients were on antidepressant | • There was a marginal effect of depression on the FIM score. |
| Morris et al 1992 | Australia | 49 | Up to 1 year | DSM III | BI | 2 patients were on antidepressant | At follow-up, people with depression Improved less (mean change from baseline, 23% versus 48%) (P = 0.001) |
| Nannetti et al. 2005 | Italy | 117 | Up to 3 months | GDS | BI | 49 patients who were depressed were on antidepressant | • Depression and functional recovery<br>• OR = 2.4 (1.1,5.1) |
| Novack et al. 1987 | USA | 134 | At discharge | SDS | BI | Not reported | • Depression was not associated with BI scores |
| Paolucci et al. 1999, | Canada | 508 | - | HDRS | BI | All depressed patients were on antidepressant | • Depression and ADL<br>• OR = 1.99 (1.14,3.46) |
| Parikh et al. 1990 | USA | 65 | Up to 2 years | HDRS | ZSRS | 2 patients were on antidepressants. | • Less impairment in ADL for non-depressed patients compared with the depressed group at 2 years follow up (t = 3.2; df = 61; p<0.01) |
| Park et al. 2015 | Korea | 180 | Up to 6 months | BDI | BI | Not reported | • Unfavorable outcome in the MSD group versus the MMD group.<br>• OR = 3.5 (1.28,9.97) |
| Park et al. 2016 | Korea | 91 | Up to 6 months | BDI | mRS GDS | Not reported | • Depression was associated with poor disability outcomes.<br>• OR = 1.37 (0.38–4.91)<br>• GDS score between MMD Vs MSD groups.<br>• Mean difference 1.0±0.2 Vs. 1.7±0.4 |
| Pellicciari et al. 2021 | Italy | 241 | - | | TCT | | • Depression and trunk control<br>  ○ $B$ = 9.057 (1.03,17.08) p = 0.027 |
| Pohjasvaara et al. 2001 | Finland | 390 | Up to 15 months | BDI | BI | 32% of depressed patients were on antidepressant | • Depression and functional outcome<br>• OR = 2.5(95% CI 1.60–3.75) |
| Saxena et al. 2007 | Singapore | 141 | Up to 6 months | GDS | BI | Not reported | • Depressive associated with low functional recovery (β = −1.31, p = 0.02) |
| Schmid et al. 2011 | USA | 367 | Up to 12 weeks | PHQ-9 | MRS | Depressed patients were on either medication or psychological care | • Baseline stroke severity and independence at 12 weeks<br>• OR = 1.06 (1.01,1.11) |
| Schubert et al. 1992 | USA | 21 | Up to 1 month | BDI | BI | Not reported | • Depression was not significantly associated with BI score changes |
| Shi et al. 2016 | China | 747 | Up to 1 year | HDRS | MRS | Not reported | • Depression and disability<br>• AOR = 4.12 (2.13,7.90) |
| Spruit-van Eijk et al. 2012 | Netherlands | 175 | At discharge | GDS | BI | Not reported | • No association between depression and BI scores |
| Tse et al. 2019 | Australia | 91 | Up to 1 year | MADRS | ACS | Not reported | • Depression and an improvement in current activity participation<br>• MADRS-SIGMA was associated with a change of 0.31 (95% CI 0.5 to 0.1, p ¼ 0.01) in current activity participation between 3- and 12 months post-stroke |
| Van de Weg et al. 1999 | Australia | 85 | Up to 6 months | GDS | FIM | 6/30 Depressed patients were on antidepressant | • Depression status and functional improvement were not associated. |
| Wagle et al. 2011 | Norway | 163 | Up to 1 year and a month | MADRS | mRS | Not reported | • MADRS were not an independent predictor of MRS scores |

(*Continued*)

**Table 2.** (Continued)

| Author, year | Setting | Sample size | Follow up | Depression measures | Outcome measure | Depression treatment | Results |
|---|---|---|---|---|---|---|---|
| Willey et al. 2010 | USA | 340 | Up to 5 years | HDRS | BI | Not reported | • Depression and severe disability at 1 year<br>• OR = 2.91 (1.07 to 7.91)<br>• Depression and severe disability at 2 years<br>• OR = 3.72 (1.29 to 10.71). |
| Wilz et al. 2007 | Germany | 81 | Up to 1 year | CDS | BI | Not reported | • Depression and functional impairment<br> ○ B = 0.17 |
| Wulsin et al 2012 | USA | 318 | Up to 1 year | CESD | MRS | Not reported | • Depression was associated with functional outcomes.<br>• OR = 2.4 (1.36–4.29) |
| Yuan et al. 2014 | China | 1753 | Up to 1 year | HDRS | MRS | 115 patients were on antidepressant | • Depression was associated with worse outcomes.<br>• OR = 1.62 (1.18–2.23) |
| Zikic et al. 2014 | Serbia | 60 | Up to 6 weeks | HDRS | BI | Not reported | • Depression and correlation between BI and HDRS score<br>• r = 0.052 (p = 0.784) |

ADL-Active Daily Living, *B*–Beta coefficient, BADL- Basic Activity of Daily Living, BI- Barthel Index, GDS–Global Deterioration Scale, HDRS- Hamilton Depression Rating Scale, MMD- Mild to Moderate Depression, MSD- Moderate to Severe Depression, OR- Odds Ratio, PSD–Post-stroke depression, SS–stroke survivor, ACS-active card sort, ADL-active daily living, BDI- beck depression inventory, BI- Barthel index, CDS- Cornell depression scale, DSM-III-R–diagnostic statistical manual of mental disorder, FIM–functional independence measure, GDS- geriatric depression scale, HADS- hospital anxiety and depression scale, HDRS–Hamilton depression rating scale, MADRS -Montgomery Rosberg depression rating scale, MBS- modified Barthel scale, MMD- mild to moderate depression, MSD- moderate to severe depression, MRS- modified Rankin scale, OR- odds ratio, PHQ-8 -patient health questionary, ZSRS- Zung self-rating scale, MMSE- Mini-mental status examination, MoCA-Montreal Cognitive Assessment, TCT- Trunk Control Test

## Impact of depression on mortality (n = 24)

Twenty-four studies [19, 69, 74–95] reported the impact of depression on the mortality rate. Of these, nine reported all-cause mortality while one study reported a suicide rate. The studies were from the USA [69, 77, 78, 83, 86, 89, 92–94], England [19, 82], Australia [74, 88], Denmark [84], Norway [90], Brazil [76], the Netherlands [79], Germany [85], Finland [87], Italy [91], Sweden and Finland [81], Lebanon [95], and South Korea [75, 80]. The sample size ranged from 84 to 152,243 participants and the age of the participants ranged from 18 to 74 years.

In the study that looked into suicide, suicide risk was higher in stroke survivors with depression compared to those without depression (AOR = 4, 95% CI (1.8–9.5)) [96]. Nineteen studies reported that depression was independently associated with an increased risk of all-cause mortality. Depression at 3 months following stroke was a predictor of mortality at 5 years of survival [19] and even over the period of 29 years [78]. Three studies reported depression was not associated with all-cause mortality (adjusted hazard ratio 1.15, 95% CI (0.76–1.75)) [69] and also no significant association between depression at baseline (one month after stroke) and long-term mortality [93].

While most of the studies did not report the treatment of depression, eight studies [80–83, 90, 91, 93, 94] reported the percentage of participants who were on antidepressant or psychological treatment. These studies showed that the probability of survival was significantly greater in the patients assigned to receive antidepressant treatment ($\chi2$ = 4.7, df = 1, p = 0.03, log-rank test) [93] and also protective (HR 0.31;95% CI 0.11 to 0.86) [94] while compared with no depression treatment group. In another study, depression was a predictor of mortality at 12 months (OR 1.1, 95% CI (0.49,2.6)) when compared to patients with depression who were given problem-solving therapy [82]. One study excluded patients who were on anti-depressant [76] (Table 3).

**Table 3. Impact of depression on mortality rate.**

| Author and year | Setting | Sample size | Age | Gender | Follow up | Measures | Depression Treatment | Results |
|---|---|---|---|---|---|---|---|---|
| Almeida & Xiao et al. 2007 | Australia | 574 | Mean 69.9 | Male 55% | Up to 10 years | ICD-9 &ICD-10 | Not reported | • Depression and mortality ○ HR = 1.26 (0.71–2.23) |
| Ayereb et al. 2014 | England | 1101 | 0-64-35.7% >64–64.3% | Male 54% | Up to 10 years | HADS | Not reported | • Depression and mortality • HR = 1.27 (1.04,1.55) |
| Boutros 2022 | Lebanon | 150 | Mean 74 | Male 58.7% | Up to a year and 3 months | HADS | Not reported | • Depression and mortality • HR = 1.30 (1.027,1.65) |
| Choi et al. 2020 | South Korea | 128,286 | Range 63–114 | Male 42.6% | U4p to 7 years | ICD-10 | Not reported | • Depression and suicide • AHR = 4.1 (1.8,9.5) |
| De Mello et al. 2016 | Brazil | 191 | Mean 63 | Male 60.2% | Up to 1 year | PHQ-9 | Patients who were on medication to treat depression were excluded. | Depression and all-cause mortality • HR = 4.60 (1.36–15.55) |
| Ellis et al. 2010 | USA | 10,025 | Range 24–74 | Male 43.1% | Up to 10 years | CES-D | Not reported | • Depression and mortality • HR = 1.88 (1.27–2.79) |
| Everson et al. 1998 | USA | 6675 | Mean 43.4 | Male 45.8% | Up to 29 years | HPL | Not reported | • Depression and mortality • HR = 1.54 (1.06,2.22) |
| Freak-Poli 2018 | Netherland | 1344 | 55+ | Male 46.1% | Up to 15 years | CES-D | Not reported | • Depression and Mortality • HR = 1.14 (1.06,1.22) |
| Hong 2018 | Korea | 210 | Mean 62.4 | Male 68.5% | Up to 8 years | ICD-10 | Documentation of antidepressants for depression after stroke were taken as PSD | • Depression and Mortality • HR = 4.93 (1.61,15.08) |
| Hornsten 2013 | Sweden and Finland | 88 | 85> | - | Up to 5 years | GDS | 28.9% of patients with depression were on antidepressant | • Depression and Mortality • HR = 1.90(1.15,3.13) |
| House et al. 2001 | UK | 448 | Median 72 | Male 54% | Up to 24 months | ICD-10 | Patients with depression were given problem-solving therapy. | • Depression and mortality • OR = 1.7 (0.95–3.0) |
| Jia et al. 2006 | USA | 5825 | Mean 67.7 | Male 98% | Up to 12 months | ICD-9 | 15% of patients with depression received antidepressant | • Crude death rate among depression 11.0% versus no depression 12.0% • OR = 0.90[0.76, 1.06] |
| Jorge et al. 2003 | USA | 104 | Range 25–84 | Male 104 | Up to 12 years | ICD-10 | 71 patients were on antidepressant | • Depression and mortality • OR = 0.74 (0.34, 1.61) |
| Jorgensen et al. 2016 | Denmark | 157,243 | < 65 24.9% | Male 23.9% | Up to 2 years | Danish Psychiatry Registry | Not reported | • Depression and all-cause mortality • HR = 1.89 (1.83,1.95) |
| Kemper et al. 2011 | Germany | 977 | >50 | Male 71% | Up to 1 year | ICD-10 | Not reported | • Depression and mortality • OR: 0.91 (0.55–1.52) |
| Melkas et al. 2010 | Finland | 257 | Mean 71 | Male 50.6% | Up to 12 years | DSM IIIR | Not reported | • Depression and mortality • HR = 1.63(1.05,2.52) |
| Morris et al. 1993 | Australia | 84 | Mean 70.8 | Male 54% | Up to 1 year and 3 months | DSM III | Not reported | • Depression and mortality • OR = 3.7 (1.1,12.2) |
| Morris et al. 1993 | USA | 91 | Mean 60.9 | Male 59% | Up to 10 years | HDRS | Not reported | • Depression and mortality • OR = 3.39 (1.4,8.4) |
| Naess et al. 2010 | Norway | 376 | Mean 72.1 | Male 60% | Up to a year | HADS | 41.3% of patients with depression were on antidepressant | • Depression and Mortality • HR = 4.4, p = 0.002 |
| Paolucci et al. 2006 | Italy | 1064 | Range 18–92 | Male 60% | Up to 2 years | BDI | 44.2 5% of patients with depression were on antidepressant | • Prevalence of mortality: 5.48% versus 4.85% (depression versus no depression • OR = 1.14 (0.65, 2.00) |

*(Continued)*

**Table 3.** (Continued)

| Author and year | Setting | Sample size | Age | Gender | Follow up | Measures | Depression Treatment | Results |
|---|---|---|---|---|---|---|---|---|
| Razmara et al. 2017 | USA | 9919 | 24–74 years | Male 60.6% | Up to 8 years | CES-D | Not reported | • Depression and mortality<br>• HR = 4.47 (1.21,16.49) |
| Ried et al. 2011 | USA | 790 | Mean 70 | Male 98% | Up to 7 years | ICD-9 | 32% of patients were on antidepressant | • Depression and mortality<br>• HR = 1.87 (1.24, 2.82) |
| Willey et al. 2010 | USA | 340 | Mean 68.8 | Male 42% | Up to 5 years | HDRS | Treatment of depression was not assessed | • Depression and all-cause mortality<br>• HR = 1.15 (0.76,1.75) |
| William et al. 2004 | USA | 51,119 | Mean 65 | Male 98% | Up to 8 years | ICD-9 | Not reported | • Depression and mortality<br>• HR = 1.13 (1.06,1.21) |

AHR-adjusted hazard ratio, CES-D- Centre for Epidemiologic Studies Depression Scale, HADS -Hospital Anxiety and Depression Scale, HPL- Human Population Laboratory, HDRS-Hamilton Depression Rating Scale, HR- Hazard Ratio, ICD-10- International Classification of Disease, PSD–Post-Stroke Depression, SS- Stroke Survivor

## Pooled impact of depression on functioning and mortality rate

Eleven studies reported the impact of depression on functioning. Stroke survivors with depression reported higher functioning problems than stroke survivors without depression (pooled OR = 1.94; 1.38, 2.73). The I-squared statistic is 85%, which tells us that 85% of the variance in observed effects reflects variance in true effects rather than sampling error (Fig 2).

Regarding the studies that report the association of depression with mortality rate, we produced two pooled estimates. The first estimate included sixteen studies that reported effect size with HR. The result showed that the mortality rate was higher among survivors with depression (pooled HR = 1.61; 1.33, 1.96) compared to survivors without depression. The I-squared statistic is 95.4%, which tells us that 95.4% of the variance in observed effects reflects variance in true effects rather than sampling error (Fig 3).

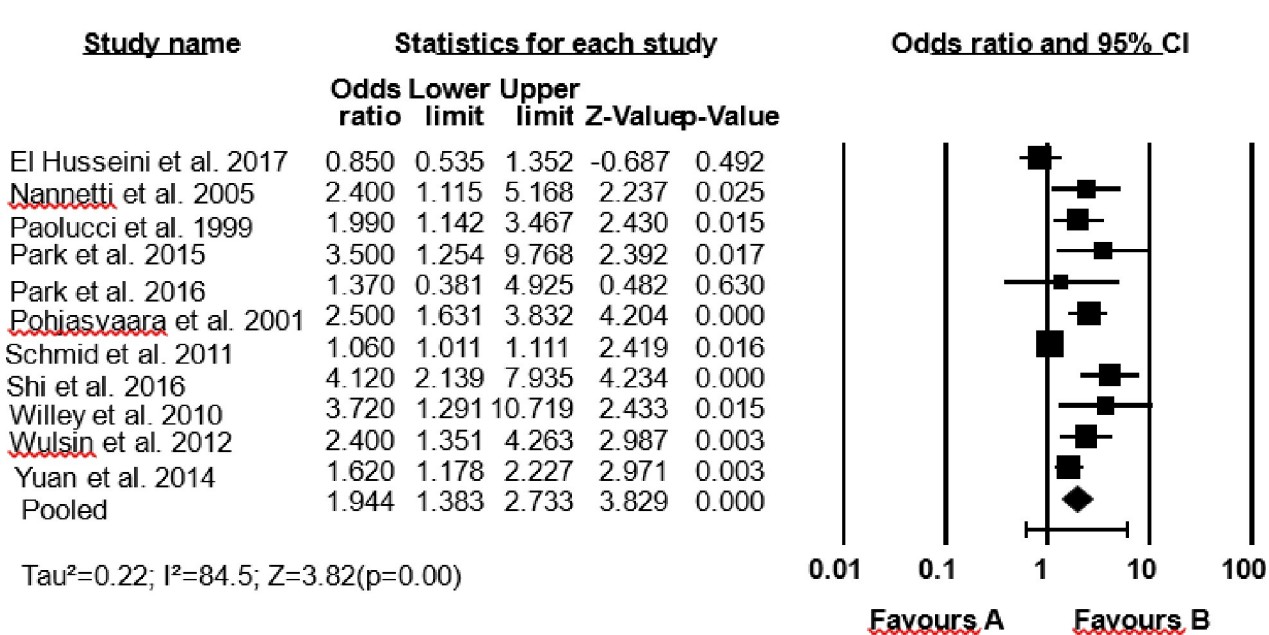

**Fig 2. A pooled estimate of the impact of depression on functioning.**

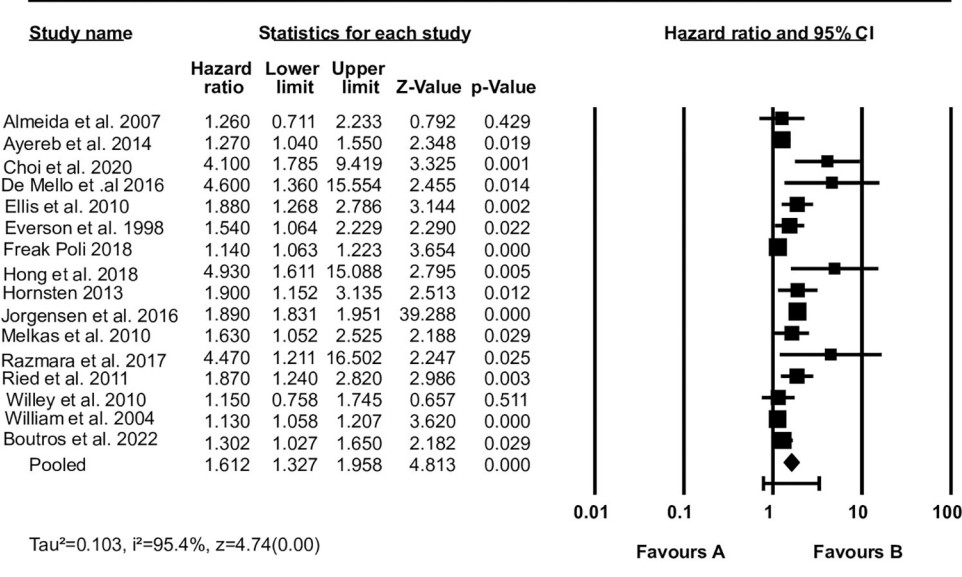

**Fig 3. A pooled estimate of the impact of depression on mortality rate (hazard ratio).**

The second estimate included seven studies that reported effect size with OR. The result showed that the mortality rate was not significantly higher among survivors with depression (pooled OR = = 1.26; 0.88, 1.80) compared to survivors without depression (Fig 4).

## Discussion

In this systematic review and meta-analysis, we have synthesized the impact of depression on stroke outcomes from the results of eighty prospective studies. Five main stroke outcomes

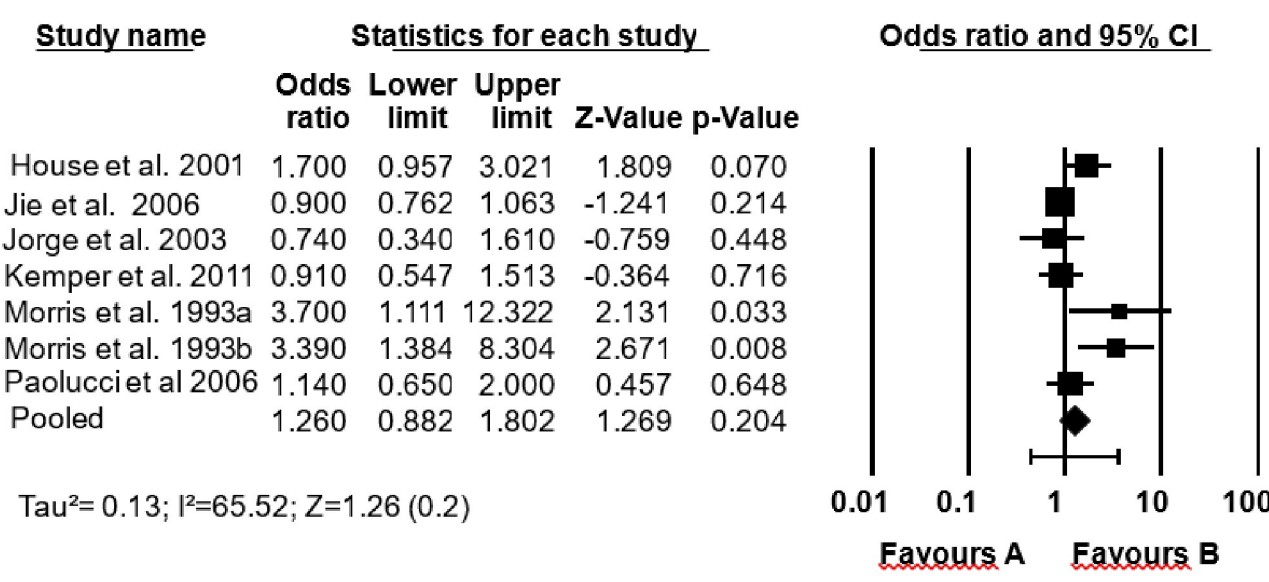

**Fig 4. A pooled estimate of the impact of depression on mortality rate (odds ratio).**

were identified: cognitive impairment, quality of life, ability to return to work, functional recovery, and mortality. We presented a comprehensive narrative report on other outcome measures and reported the pooled impact of depression on functional recovery and mortality rate which were not fully addressed in previous reviews.

Three out of the four studies that investigated the impact of depression on the ability to return to work [15–17] reported that depression was not related to the stroke survivor's ability to return to work. This finding may not be true as the survivors in these studies had mild to moderate stroke severity, and they were in the 'younger' age group. Other studies also refuted this conclusion [14, 16]. The inability to return to work is also associated with other factors such as cognitive status [17] and health insurance [14]. The inability to return to work may be different across different settings. These included the socioeconomic status and the available social welfare system which warrants the need for primary studies in low-resource settings to fully understand the impact and find ways to address these issues.

Our finding on the impact of depression on cognitive impairment is not to our expectation [19, 32, 35]. The no association result may be due to lower stroke severity and the exclusion of people with severe cognitive impairment from the studies. Previous studies reported cognitive impairment among both people with depression [97] and stroke [98]. We expect a multiplicative effect when a person had both depression and stroke. However, the results of these studies show the need for more studies to establish the relationship between depression and cognition among stroke survivors.

The association between depression and quality of life after stroke seems conclusive though two out of the twelve studies reported no association in a multivariable analysis. This might be because of the low rate of depression in those studies, and study participants had mild to moderate stroke severity. In these studies, stroke survivors who received treatment for depression were not included in the analysis [22, 29]. Patients who were treated in outpatient clinics and aphasic stroke survivors were also not included in the studies.

Regarding functional recovery, most of the studies indicated the impact of depression on this outcome while few studies reported low to no impact of depression on functional recovery. In these studies, participants who received immediate pharmacological treatment for depression showed better functional improvement by 30% compared to those who did not get treatment for depression [99]. This was also supported by our meta-analysis where depression was significantly associated with poorer functioning though the result needs to be interpreted with caution since the heterogeneity among studies is high.

Our pooled estimate of the impact of depression on mortality rate had significant implications for policy and practice, as shown in previous meta-analyses [6, 7, 9]. Depression increases both all causes, and suicide-related death compared to those without depression. Studies also suggested that early intervention with antidepressant treatments could be associated with the probability of longer survival [93, 94].

Though our review can be considered a comprehensive systematic review with a meta-analysis which synthesized a significantly higher number of prospective studies compared to previous reviews, it is not free from limitations. The first limitation was the definition of post-stroke depression. Stroke is mainly associated with chronic health conditions such as hypertension and diabetes. Depression is a common health problem among people with these chronic health conditions [100, 101]. These indicated the depression after stroke may be a continuation of pre-stroke depression. Nevertheless, the data shows that depression remains a significant issue that needs to be addressed for all stroke patients regardless of when it occurs. The second limitation was the heterogeneity of the studies. The heterogeneity was related to illness duration, the severity of the stroke, the setting and the measures used to assess depression. It's important to consider all these factors while reading the finding.

The third limitation was related to the small sample size of the studies included in the meta-analysis which prevented us from sub-group analysis. The fourth limitation comes from the fact that most of the studies were conducted in high-income countries with better rehabilitation services.

## Conclusion

Depression affects many aspects of stroke outcomes including survival. The evidence is not conclusive on some outcomes such as cognition. This review indicated the need for longitudinal studies with higher sample size especially in low-resource settings since the treatment for depression and stroke is not well-established and there is no well-established social welfare system. It also showed the need to provide mental health support for stroke survivors as it is related to better overall health and recovery.

## Supporting information

**S1 File. Search strategy for impact of depression on stroke outcomes.**
(DOCX)

**S2 File.**
(PDF)

**S1 Table. Quality assessment for impact of depression on stroke outcomes.**
(DOCX)

## Author Contributions

**Conceptualization:** Seble Shewangizaw, Atalay Alem.

**Data curation:** Seble Shewangizaw, Wubalem Fekadu.

**Investigation:** Seble Shewangizaw.

**Methodology:** Seble Shewangizaw, Wubalem Fekadu, Atalay Alem.

**Software:** Seble Shewangizaw, Wubalem Fekadu.

**Supervision:** Catherine Sackley, Atalay Alem.

**Writing – original draft:** Seble Shewangizaw.

**Writing – review & editing:** Seble Shewangizaw, Wubalem Fekadu, Yohannes Gebregzihabhier, Awoke Mihretu, Catherine Sackley, Atalay Alem.

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
