## [Decision Letter · Decision Letter 0]

26 Jun 2023

PONE-D-23-07673Impact of depression on stroke outcomes among stroke survivors: Systematic review and meta-analysis.PLOS ONE

Dear Seble Shewangizaw**,**

Thank you for submitting your manuscript to PLOS ONE. After careful consideration, we feel that it has merit but does not fully meet PLOS ONE’s publication criteria as it currently stands. Therefore, we invite you to submit a revised version of the manuscript that addresses the points raised during the review process.

Please submit your revised manuscript by July 26,2023 11:59 PM. If you will need more time than this to complete your revisions, please reply to this message or contact the journal office at plosone@plos.org. Please include the following items when submitting your revised manuscript:A rebuttal letter that responds to each point raised by the academic editor and reviewer(s). You should upload this letter as a separate file labeled 'Response to Reviewers'.A marked-up copy of your manuscript that highlights changes made to the original version. You should upload this as a separate file labeled 'Revised Manuscript with Track Changes'.An unmarked version of your revised paper without tracked changes. You should upload this as a separate file labeled 'Manuscript'.

We look forward to receiving your revised manuscript.

Kind regards,

Saraswati Dhungana, MD

Academic Editor

PLOS ONE

Journal Requirements:

“This work was supported through the DELTAS Africa Initiative (DEL-15-01). The DELTAS Africa Initiative is an independent funding scheme of the African Academy of Sciences (AAS) Alliance for Accelerating Excellence in Science in Africa and supported by the New Partnership for Africa’s Development Planning and Coordinating Agency (NEPAD Agency) with funding from the Wellcome Trust (DEL-15-01) and the UK government. The views expressed in this publication are those of the author(s) and not necessarily those of AAS, NEPAD Agency, WellcomeTrust or the UK government.”

Reviewers' comments:

Reviewer's Responses to Questions

**Comments to the Author**

1. Is the manuscript technically sound, and do the data support the conclusions?

Reviewer #1: No

2. Has the statistical analysis been performed appropriately and rigorously? 

Reviewer #1: No

3. Have the authors made all data underlying the findings in their manuscript fully available?

Reviewer #1: Yes

4. Is the manuscript presented in an intelligible fashion and written in standard English?

Reviewer #1: No

5. Review Comments to the Author

Reviewer #1: This study investigated the impact of depression on stroke outcome which is very important area of research among stoke survivors. I have added my comments for the improvement of quality of the paper.

Title and abstract looks fine. Introduction is presented well with highlighting the gap in literature and there is clarity of research questions.

Methods

Eligibility criteria: It is clear however it’s better to present in sentences( paragraph) than in a bullet format.

Data analysis: There are some grammatical errors in writing, Please use past tense. Please mentioned the methods of meta-analysis clearly here and no need of repeating same in the result sections.

Study selection process: Please clarify whether you used endnote for the whole screening process or any other SR software such as Revman or Covidence was also used.

Result

PRISMA flow diagram It can be presented in better ways. Please refer this http://prisma-statement.org/prismastatement/flowdiagram.aspx?AspxAutoDetectCookieSupport=1 and update the diagram accordingly. In addition, I would suggest witting excluded heading above excluded study.

In addition, I can see reasons of exclusion after full text review are only due to the wrong study design; however usually there would be chances of having other reasons ( wrong study population, wrong outcomes etc.) So please make sure and report if there are any other reasons of exclusion as well.

Characteristic of the study

Please keep the reference of each studies after description such as most of the studies (66 from high income countries ) USA ( keep ref) , England(ref) ……similarly keep reference of the each included study immediately after their description so that reader can locate them easily.

Same comments applies to other places as well such as 4 studies assessed the PSD – keep ref for those study immediately after text. Same goes to - Risk of Bias section - keep references of studies with strong, moderate and weak quality.

Inside table 1 as well , keep ref after each study. The length of follow up in each studies varies significantly such as from 28 days up to 10 year, I was wondering whether such variation of study duration has some effects on the outcome or not? why the length of follow up was not considered during deciding the inclusion criteria of the studies? Please briefly clarify your approach.

Same comments, author has not cited the references properly to locate the study in most of the places such as second para under impact of depression on cognitions.

Please apply above comment in each places whenever you refer any included studies, keep the reference immediately after that to locate them easily.

Language editing of the manuscript is advised.

Meta-analysis

The method of meta analysis can be included in the method section than in the result. Authors has not interpreted the meta analysis finding properly for example in text of Fig 2 only. The effect size was 1.94 with a 95% confidence interval of 1.38 to 2.73 is mentioned. But you should report what does 1.94 mean in this case so that reader can get the result correctly and similarly interpret and report each forest plot finding accordingly.

I would suggest writing the descriptive result concisely than repeating and putting findings here and there. For e.g., descriptive result and meta-analysis of same result ( such as PSD and cognition) can be kept together and interpret accordingly rather than keeping all the forest plot separately in the text. In addition, range of OR in forest plot is very wide 1, 10 and 100… please try to revise it.

In addition, significance of the findings presented in the forest plot is not interpreted these finding accurately for e.g. what is the difference in finding of fig 1-3 and 4, what does those non-significant findings of fig 4 and significance in other figures means. Please take care of everything and report them clearly.

In addition, what about the quality of all the included studies? whether some of the studies with strong evidence and some with weak were analyzed together or not in meta-analysis ? as only homogenous study should be included in the meta-analysis. If that has any effect in the pooled result or not. Please clarify

Discussion

This section needs major revision. The comparison with previous evidences and critical interpretation and argument is lacking in this section. I would advise rather than repeating what is mention in the result section, please try to interpret the findings and present possible reasons and ways forward with the support of evidences.

In addition, there is plenty of grammatical and typos errors throughout the manuscript. Even the spelling of meta-analysis is spelled as "metanalysis" in multiple places. Please double check.

Conclusion

Conclusion is not presented strongly. Please don’t write basic and general conclusion, it should be based on your study findings.

Usually SRM evidence are considered strong for the future policy and program but I can’t see any special recommendation from this SRM. Please clarify.

Overall good attempt by the authors. But this manuscript need major language editing and revision of content and process to improve the quality of the paper.

Best wishes

6. PLOS authors have the option to publish the peer review history of their article (what does this mean?). If published, this will include your full peer review and any attached files.

Reviewer #1: **Yes: **Buna Bhandari

Reviewer #2 Comments to the authors:

The authors present a systematic review and meta- analysis on “Impact of depression on stroke outcomes among stroke survivors.” This is an important area and I congratulate the authors on coming up with this piece of writing. I advise the authors to further update on if any further reviews or prospective studies have been conducted meanwhile.

I have the following comments on the manuscript.

Title: The title appears misleading if the authors are looking at the effects of post- stroke depression. Depression in stroke patients does not always appear post- stroke. Those might have had depressive diagnosis before having stroke. So, I request them to clarify. Please clarify this issue throughout the manuscript at all instances.

Introduction: The second sentence requires paraphrasing in the first paragraph. In second paragraph, please be specific regarding post- stroke depression. When you say post-stroke depression, do you have any operational definition for post stroke depression? Please clarify because this is the main theme for your SR. Also, mention briefly why post- stroke depression is important from theoretical perspective and its clinical relevance. You can also give prevalence estimates from updated reviews. In the background information where the authors cite multiple reviews, what do those reviews tell about the effects of post stroke depression? Do all the reviews mention regarding post stroke depression in stroke survivors?

Search strategies: I wonder why the authors tried to look into grey literature when their search eligibility is limited to the four listed database. Also, please review if you meant supplementary file 1 and what does that correspond to.

Eligibility criteria: What is the operational definition of post stroke depression? There are many outcomes in the study, and they are quite broad. Could this be a limitation to the study? For example, when you say cognition, it denotes many things. What were the authors referring to when they mention cognition?

Did you have any exclusion criteria? Did you come across situations when it was tough to decide to include or exclude a study based on your eligibility?

Study selection process: Who was the third reviewer involved in discrepancy resolving? Was there any discrepancy?

The authors do not mention if risk of bias assessment was done though quality check was done. Please clarify and what tool was used? To me, it appears there was overlap between he two terms in the manuscript.

Results: Tables

I wonder if all the studies included are prospective. There is great heterogeneity in the studies included. I also see from table 1 that in some studies, e.g., Gbiri et al. 2010., depression at onset is mentioned. What does this mean? In another study by Guajardo et.al. 2015. and Donellan et al. 2010. , there is mention of depressive symptoms, rather than diagnosis of depression. Depressive symptoms are common in acute stages post-depression, but depression diagnosis deserves clinical attention for various reasons. Please clarify.

The authors are further advised to mention the setting of the studies besides the country such as outpatient, inpatient, physiotherapy etc. and also the title of the studies in the table itself.

Same goes for other tables. Please clarify.

Meta- analysis: Please provide a clear rationale for inclusion of specific studies. What was your aim to do a meta- analysis before planning this? The authors are also advised to provide inferences about the reported results. Did you measure variance? Please explain.

There are other issues as well. Some methodology statements are described in results section. Please consider keeping them where suited best.

Discussion: This section needs major rewriting based on all the clarifications after comments. Further, the authors are advised to discuss based on the reported findings and literature available. I believe there are other biases and limitations besides those mentioned. Please elaborate on those. 

Minor issues:

There are many grammatical and typo errors throughout the manuscript. Also, I see that the supplementary files are not correctly named and not placed sequentially in the texts. Please correct them according to journal’s policy.

All the best.

---

## [Author Response · Author response to Decision Letter 0]

11 Aug 2023

We appreciate the comments and suggestions of all the reviewers and editorial requests. We have tried to address the concerns of each reviewer and the editorial requests in the following table.

---

## [Decision Letter · Decision Letter 1]

5 Sep 2023

PONE-D-23-07673R1Impact of depression on stroke outcomes among stroke survivors:Systematic review and meta-analysis.PLOS ONE

Dear Dr. Shewangizaw,

Thank you for submitting your manuscript to PLOS ONE. After careful consideration, we feel that it has merit but does not fully meet PLOS ONE’s publication criteria as it currently stands. Therefore, we invite you to submit a revised version of the manuscript that addresses the points raised during the review process.

We look forward to receiving your revised manuscript.

Kind regards,

Saraswati Dhungana, MD

Academic Editor

PLOS ONE

Journal Requirements:

**Additional Editor Comments:**

Please respond to all of the reviewer's comments. Otherwise, the manuscript looks good to me.

Reviewers' comments:

Reviewer's Responses to Questions

**Comments to the Author**

1. If the authors have adequately addressed your comments raised in a previous round of review and you feel that this manuscript is now acceptable for publication, you may indicate that here to bypass the “Comments to the Author” section, enter your conflict of interest statement in the “Confidential to Editor” section, and submit your "Accept" recommendation.

Reviewer #1: (No Response)

2. Is the manuscript technically sound, and do the data support the conclusions?

Reviewer #1: Yes

3. Has the statistical analysis been performed appropriately and rigorously? 

Reviewer #1: Yes

4. Have the authors made all data underlying the findings in their manuscript fully available?

Reviewer #1: Yes

5. Is the manuscript presented in an intelligible fashion and written in standard English?

Reviewer #1: Yes

6. Review Comments to the Author

Reviewer #1: Author has addressed most of the previous comments made by reviewers. Quality of paper is improved than before. However, there are minor suggestions for making it publishable

1. Abstract- data extraction- please write two authors extracted data from the studies than writing "extracted the studies"

2. You can provide PROSPERO registration number below abstract than within abstract.

3. Better to write abstract accordingly to PLOS one Standard format - Intro, Methods, Results and conclusions

4. Result section stills lacks adequate referencing to the respective articles especially in the characteristics of the included study and ROB section. It would be difficult to locate included studies if it is not clearly referred within text.

5. In addition, number of the articles are edited in the revised PRISMA diagram, I was wondering about the reason. I assume author did not run search again after submission of the first manuscript. In addition, it is not appropriate to write hand search - 2 in the full text review section in the PRISMA diagram. Was those 2 article were directly included for full text review without doing title and abstract screening. Not clear to me.

6. I still preferred to have different heading for the result of meta-analysis rather than keeping meta-analysis as a subheading. It is a statistical methods not the content of result.

7. Discussion and conclusion are improved than before.

8. References are not uniform fonts style as manuscript. Please follow the journal guidelines for formatting the article.

Best wishes

7. PLOS authors have the option to publish the peer review history of their article (what does this mean?). If published, this will include your full peer review and any attached files.

Reviewer #1: No

---

## [Author Response · Author response to Decision Letter 1]

25 Sep 2023

We appreciate the reviewers and the editor for the comments and suggestions and for recognizing the updates we made. We have tried to address the current concerns and the editorial issues. Please see the details in the following table and the track changes in the manuscript.

---

## [Decision Letter · Decision Letter 2]

11 Oct 2023

PONE-D-23-07673R2Impact of Depression on Stroke Outcomes among Stroke Survivors: Systematic review and Meta-analysis.PLOS ONE

Dear Dr. Shewangizaw,

Thank you for submitting your manuscript to PLOS ONE. After careful consideration, we feel that it has merit but does not fully meet PLOS ONE’s publication criteria as it currently stands. Therefore, we invite you to submit a revised version of the manuscript that addresses the points raised during the review process.

We look forward to receiving your revised manuscript.

Kind regards,

Saraswati Dhungana, MD

Academic Editor

PLOS ONE

Journal Requirements:

Additional Editor Comments :

Based on the review from a reviewer, please address the comments.

Best

Saraswati Dhungana

Academic Editor

Reviewers' comments:

Reviewer's Responses to Questions

**Comments to the Author**

1. If the authors have adequately addressed your comments raised in a previous round of review and you feel that this manuscript is now acceptable for publication, you may indicate that here to bypass the “Comments to the Author” section, enter your conflict of interest statement in the “Confidential to Editor” section, and submit your "Accept" recommendation.

Reviewer #1: All comments have been addressed

2. Is the manuscript technically sound, and do the data support the conclusions?

Reviewer #1: Yes

3. Has the statistical analysis been performed appropriately and rigorously? 

Reviewer #1: Yes

4. Have the authors made all data underlying the findings in their manuscript fully available?

Reviewer #1: Yes

5. Is the manuscript presented in an intelligible fashion and written in standard English?

Reviewer #1: Yes

6. Review Comments to the Author

Reviewer #1: Thank you for addressing most of the comments. Manuscript is improved alot.

However, I still have some concern.

Format of abstract- Though it is systematic review, without background (showing gap in the literature), directly keeping objectives does not provide enough information for the reader.

See previous published SRM of PLOS One and follow guidelines accordingly

https://journals.plos.org/plosone/article?id=10.1371/journal.pone.0150625

In addition, data source, eligibility, data analysis all comes under the methods section of SRM. so it is advisable to incldue them under methods.

Secondly, I still don't see significance of keeping Hand search (2 article) in full text section directly in PRISMA flow diagram. Because, if your search startegy is not comprehensive and could not search any previously published studies then hand-search is a way to ensure all the eligible study is included in your Review.

So, to decide whether cross references will be included in the study or not, first reviewer should do title and abstract review of cross references and then only includes them into full text review; not directly to full text review based on the SRM guidelines so it is suggested to include those 2 hand-search article under title and abstract screen section of PRISMA diagram than directly to the full text screen.

However, editor can take a final call on it.

Best wishes

7. PLOS authors have the option to publish the peer review history of their article (what does this mean?). If published, this will include your full peer review and any attached files.

Reviewer #1: No

---

## [Author Response · Author response to Decision Letter 2]

2 Nov 2023

We appreciate the reviewer and the editor for the comments and suggestions and for recognizing the updates we made. We have tried to address the current concerns and the editorial issues. Please see the details in the following table and the track changes in the manuscript.

---

## [Decision Letter · Decision Letter 3]

7 Nov 2023

Impact of Depression on Stroke Outcomes among Stroke Survivors: Systematic review and Meta-analysis.

PONE-D-23-07673R3

Dear Mrs. Seble Shewangizaw

We’re pleased to inform you that your manuscript has been judged scientifically suitable for publication and will be formally accepted for publication once it meets all outstanding technical requirements.

Kind regards,

Saraswati Dhungana, MD

Academic Editor

PLOS ONE

Additional Editor Comments (optional):

Reviewers' comments:

Reviewer's Responses to Questions

**Comments to the Author**

1. If the authors have adequately addressed your comments raised in a previous round of review and you feel that this manuscript is now acceptable for publication, you may indicate that here to bypass the “Comments to the Author” section, enter your conflict of interest statement in the “Confidential to Editor” section, and submit your "Accept" recommendation.

Reviewer #1: All comments have been addressed

2. Is the manuscript technically sound, and do the data support the conclusions?

Reviewer #1: Yes

3. Has the statistical analysis been performed appropriately and rigorously? 

Reviewer #1: Yes

4. Have the authors made all data underlying the findings in their manuscript fully available?

Reviewer #1: Yes

5. Is the manuscript presented in an intelligible fashion and written in standard English?

Reviewer #1: Yes

6. Review Comments to the Author

Reviewer #1: (No Response)

7. PLOS authors have the option to publish the peer review history of their article (what does this mean?). If published, this will include your full peer review and any attached files.

Reviewer #1: **Yes: **Buna Bhandari

---

## [Editor Report · Acceptance letter]

16 Nov 2023

PONE-D-23-07673R3 

Impact of Depression on Stroke Outcomes among Stroke Survivors: Systematic review and Meta-analysis. 

Dear Dr. Shewangizaw:

I'm pleased to inform you that your manuscript has been deemed suitable for publication in PLOS ONE. Congratulations! Your manuscript is now with our production department. 

Kind regards, 

on behalf of

Dr. Saraswati Dhungana 

Academic Editor

PLOS ONE